# Effects of Pre-Emulsification with Thermal-Denatured Whey Protein on Texture and Microstructure of Reduced-Sodium Processed Cheese

**DOI:** 10.3390/foods12152884

**Published:** 2023-07-29

**Authors:** Hongjuan Li, Yumeng Zhang, Hongyu Cao, Yuchen Zhang, Junna Wang, Yumeng Zhang, Xiaoyang Pang, Jiaping Lv, Shuwen Zhang, Jinghua Yu

**Affiliations:** 1State Key Laboratory of Food Nutrition and Safety, Key Laboratory of Food Nutrition and Safety, Ministry of Education, College of Food Science and Engineering, Tianjin University of Science and Technology, No. 29, No. 13 Ave., TEDA, Tianjin 300457, China; 2Institute of Food Science and Technology, Chinese Academy of Agricultural Sciences, Beijing 100193, Chinalvjp586@vip.sina.com (J.L.)

**Keywords:** reduced sodium, processed cheese, emulsifying salt, whey protein, pre-emulsification

## Abstract

Thermal-denatured whey protein-milk fat emulsion gels with different degrees of pre-emulsification were prepared by pre-emulsifying milk fat with thermal-denatured whey protein and used in the preparation of reduced-sodium processed cheeses. The effect of the thermal-denatured whey protein pre-emulsification process on the texture and microstructure of reduced-sodium processed cheeses was evaluated by studying the composition, color, texture, functional properties, microstructure and sensory analysis of the processed cheeses. The results showed that compared with cheese without pre-emulsified fat (1.5% ES control), the moisture content of cheese with pre-emulsified 100% fat (1.5% ES100) increased by 5.81%, the L* values increased by 7.61%, the hardness increased by 43.24%, and the free oil release decreased by 38%. The microstructure showed that the particle size of fat was significantly reduced, and the distribution was more uniform. In addition, compared with the cheese added with 3% emulsifying salt (3% ES control), the amount of emulsifying salt in the 1.5% ES100 decreased by 50%, but the fat distribution of the two kinds of cheese tended to be consistent, and there was no obvious change in texture characteristics and meltability. Sensory scores increased with the increase in pre-emulsification degree. Overall, the pre-emulsification of milk fat with thermal-denatured whey protein can reduce the sodium content of processed cheese and improve its quality.

## 1. Introduction

Processed cheese, which is rich in nutritional value, is a homogeneous product formed by heating and continuous mixing of one or two natural cheeses, water, emulsifying salts and other dairy or non-dairy products [1]. It has a high level of acceptance due to its wide range of sources, low fermentation flavor and long shelf life [2]. However, the amount of sodium in processed cheese is usually higher (650–1600 mg 100 g^−1^) than that in natural cheese (190–1400 mg 100 g^−1^) [3]. Numerous studies have shown that high sodium intake reduces the absorption of calcium in the body and can also lead to heart disease, hypertension and stroke [4]. Therefore, reduced-sodium processed cheese has great development prospects.

The 28–37% of sodium in processed cheese comes from natural cheese used as a raw material, and 44–48% comes from emulsifying salts used for processed cheese production. Emulsifying salts are essential auxiliary materials in the processing of processed cheese, which have the functions of adjusting pH and ion exchange, promoting protein hydration and fat emulsification [5]. Therefore, emulsifying salts are essential for the processing of processed cheese and how to reduce the amount of emulsifying salts in processed cheese is an important problem to be solved [6]. Compared with traditional processed cheese, reducing the emulsifying salt concentration by up to 15% can produce a cheese with similar functional characteristics [3]. El-Bakry, et al. [7] found that the hardness of the cheese increased and the meltability decreased as the emulsifying salt was reduced. The maximum reduction in emulsifying salt was 40%, but this resulted in long processing times, which affected the stability and functional properties of the product. Therefore, directly reducing the level of emulsifying salts in processed cheese not only affects the stability of the cheese but also alters the flavor and texture of the cheese. Both sodium and potassium ions are able to displace calcium ions from the original cheese, thus restoring good hydration and emulsification of the casein, but processed cheese produced by sodium ions usually has better functional and sensory characteristics than processed cheese produced by potassium ions [8]. Thus, potassium salt cannot completely replace sodium salt as an emulsifier, and only partial substitutions can be made. However, the processed cheese made by these methods still has some defects in flavor and texture. Ehsannia and Sanjabi [9] partially replaced the sodium salt with potassium salt and they found that as the storage period increased, the bitterness became more pronounced, and the flavor was significantly reduced in reduced-sodium cheeses compared to full-sodium cheeses.

At present, researchers have replaced sodium-containing emulsifying salts with substances with similar functional properties to those of emulsifying salts, such as proteins and polysaccharides [10]. Whey proteins have good water absorption and emulsification properties like those of emulsifying salts [11]. Sołowiej, et al. [12] used polymerized whey protein isolate (0.2 to 0.4% WPI) as a partial replacement for emulsifying salt in the production of processed cheese. It was found that the tested samples all had good meltability. Protein-based emulsion systems assembled entirely from natural ingredients have been used to improve flavor, protect active ingredients and increase bioavailability [13,14]. Whey protein emulsion gels are usually a semi-solid material, and emulsion gels made with whey protein pre-emulsified oils can be used to make new food products or to improve the texture of products such as yogurt and cheese [11]. Li, et al. [15] found that pre-emulsification of milk fat using thermal-denatured whey proteins to make emulsion gels, which were added to low-fat yoghurt, significantly improved the textural properties and water-holding capacity of the samples. In a previous study, we also found that processed cheeses made with thermal-denatured whey protein emulsion gels had a more homogeneous fat distribution [16].

Therefore, in this study, thermal-denatured whey protein-milk fat emulsion gels with different degrees of pre-emulsification were prepared using thermal-denatured whey protein to pre-emulsify milk fat and used in the preparation of reduced-sodium processed cheeses. The effects of the thermal-denatured whey protein pre-emulsification process on the texture, functional properties and microstructure of reduced-sodium processed cheese were further investigated. Studies were also carried out on processed cheeses with different levels of emulsification salt content and degrees of pre-emulsification. This study provides a new approach to the processing of reduced-sodium processed cheeses.

## 2. Material & Methods

### 2.1. Materials and Chemicals

Whey protein isolate (WPI, 89 wt% protein) was produced by Hilmar Ingredients (Hilmar, CA, USA). Rennet casein (82 wt% protein) was supplied by Fonterra Co-operative Group (Auckland, New Zealand). Natural mozzarella cheese (23.0 wt% protein, 23.0 wt% fat, 393 mg/100 g Na) was supplied by Mubao Food Technology Co., Ltd. (Shanghai, China). Anhydrous milk fat (99.8 wt% fat) was purchased from Mubao Food Technology Co., Ltd. (Shanghai, China). Citric acid was purchased from Youbaojia Food Co., Ltd. (Henan, China). Sodium citrate and sodium pyrophosphate were purchased from Shenzhen Xingmu Bioengineering Co., Ltd. (Shenzhen, China).

### 2.2. Processed Cheese Preparation

#### 2.2.1. Pre-Emulsification of Milk Fat with Thermal-Denatured Whey Protein

It was prepared according to the previously reported method [15]. WPI was dissolved in distilled water and rapidly stirred with a high-speed disperser (FJ200-SH, Shanghai Standard Model Co., Ltd., Shanghai, China) at 9000 rpm for 3 min at room temperature, followed by hydration in a water bath at 55 °C for 1 h and finally denaturation at 90 °C for 20 min to obtain WPI dispersions with a mass fraction of 5%. Then, milk fat (60 °C) was added to the WPI solution and stirred rapidly at 10,000 rpm for 3 min with a high-speed disperser. After cooling to room temperature, the pH of the emulsion was adjusted with citric acid to 6.0. After storage at 4 °C for 12 h, thermal-denatured whey protein-milk fat emulsion gels (WPI-EGs) were formed.

#### 2.2.2. Processed Cheese

The cheese processing was carried out according to our previous study [16]. Seven formulations were prepared by adding emulsion gels with different contents of emulsifying salts and different degrees of pre-emulsification as follows (Table 1): (1% ES100) = experimental group added with 1 wt% emulsifying salt and pre-emulsified 100% fat in the cheese formulation with 5 wt% WPI dispersion, (1.5% ES control) = control group added with 1.5 wt% emulsifying salt without pre-emulsification process, (1.5% ES25) = experimental group added with 1.5% emulsifying salt and pre-emulsified a quarter fat in the cheese formulation with 5 wt% WPI dispersion, (1.5% ES50) = experimental group added with 1.5% emulsifying salt and pre-emulsified a half fat in the cheese formulation with 5 wt% WPI dispersion, (1.5% ES100) = experimental group added with 1.5% emulsifying salt and pre-emulsified 100% fat in the cheese formulation with 5% WPI dispersion, (3% ES control) = control group added with 3% emulsifying salt without pre-emulsification process and (3% ES100) = experimental group added with 3% emulsifying salt and pre-emulsified 100% fat in the cheese formulation with 5 wt% WPI dispersion.

The mixture of water, milk fat, natural cheese, casein, dissolved emulsified salt, WPI-EGs or fat was placed in vacuum homogenizing emulsifying machine (ZJR-5, Wuxi Yikai Automation Technology Co., Ltd., Wuxi, China) and the mixture was stirred at a speed of 300 r/min at −0.1 MPa. After 40 min, the temperature was raised from room temperature to 90 °C and held for 4 min. Each sample was produced in duplicate.

### 2.3. Cheese Composition

Moisture, protein, fat, pH and fiber were determined as described previously [17]. The sodium content was determined according to the method of Ferrão, et al. [18] by inductively coupled plasma mass spectrometry ICAPQ (Thermo Fisher, Karlsruhe, Germany).

### 2.4. Color Evaluation

Color evaluation of the samples was carried out according to the method of Aktypis, et al. [19] using a high quality colorimeter NR110 (Shenzhen sanenshi Technology Co., Ltd., Shenzhen, China). L* (brightness), a* (red value and green value) and b* (yellow value and blue value) were used to indicate the color change degree of the sample. Each sample was measured in triplicate.

### 2.5. Texture Profile Analysis (TPA)

The method of Li, Qin, Yu, Han, Zheng, Li and Yu [16] was referred to with some modifications. A 15 × 15 × 15 mm cube sample was cut from the center of the processed cheese. TPA analyses were performed on cheeses after 2 weeks of storage at 4 °C using a TA-XT2i (Stable Micro Systems, Surrey, UK). The parameters of texture analyzer were set as follows: before the test, the probe descending speed was 4.0 mm/s; the test speed was 1.0 mm/s; after the test, the probe return speed was 1.0 mm/s; the compression deformation was 30%; the trigger force was 10 g; and the test was performed using a cylindrical metal compression plate of 36 mm diameter. All tests were performed at room temperature, and all analyses were performed in triplicate.

### 2.6. Melt and Free Oil Release Test

Melt and free oil release tests were performed according to our previous method with modifications [16]. The cheese was cut into cylinders of 7 mm thickness and 15 mm diameter. For the melt test, the samples were heated in glass test tubes at 100 °C for 1 h. After cooling for 30 min, the melt diameter of the cheese product in the test tubes was measured, and the flow length represented its melt characteristics. To determine the release of oil, the cheese was placed in a petri dish with filter paper and heated at 100 °C for 1 h. After cooling for 30 min, the diameter of the formed free oil ring was measured in four directions, then the average values were calculated. Each measurement was repeated three times.

### 2.7. Confocal Laser Scanning Microscopy (CLSM)

Cheese samples were visualized by confocal laser scanning microscopy (CLSM) (TCS SP5II, Leica Microsystems Co., Ltd., Hessian, Germany) according to Žolnere, et al. [20]. The cheese was cut into 5 mm × 5 mm × 1 mm slices. The sections were first immersed in 1 mg/mL Fast Green solution, and after 5 min, the stain was poured out and rinsed three times with phosphate-buffered saline (pH 7.0). Subsequently, the sections were immersed in 0.1 mg/mL ethanol solution of Nile Red, and after 5 min, the stain was poured out and rinsed three times with phosphate-buffered saline (pH 7.0). The excitation wavelengths of Nile Red and Fast Green are 488 nm and 633 nm, respectively, and the emission wavelengths are 500–580 nm and 650–700 nm, respectively.

### 2.8. Sensory Analysis

Twenty people trained in the sensory evaluation of cheese products were recruited as evaluators (10 women and 10 men, average age: 25 years old). During training, different cheese samples were provided to help group members identify the appearance, color, texture, aroma and firmness attributes of cheese. The 10-point intensity scale was used for each term, with 1 being the lowest score and 10 the maximum. The study was reviewed and approved by the Tianjin University of Science and Technology IRB and informed consent was obtained from each subject prior to their participation in the study.

### 2.9. Statistical Analysis

The data was analyzed using SPSS 13.0 (SPSS Inc., Chicago, IL, USA). Analysis of variance (ANOVA, *p* < 0.05) with Tukey’s test were used to test the significant differences of samples. Each measurement was expressed as the mean ± standard deviation.

## 3. Results and Discussion

### 3.1. Cheese Composition

The cheese composition is shown in Table 2. There was no significant difference (*p* > 0.05) in protein and fat contents of all samples; the results showed that the amount of emulsifying salt and the degree of pre-emulsification had no effect on the protein and fat content of the sample. The 1% ES100 group had the lowest Na content (271.06 mg/100g). The 3% ES100 group had the highest Na content (947.08 mg/100g). In the 1.5% ES group, the total Na content was reduced by 40–45% compared with the 3% ES group. The ash content, pH value and sodium content increased with the increase in emulsifying salt. The 3% ES control group had the lowest moisture content (53.30 g/100 g), whereas in the 1.5% ES group (1.5% ES control, 1.5% ES25, 1.5% ES50, 1.5% ES100), the moisture content of the cheese samples increased significantly (*p* < 0.05) with the increase in the pre-emulsified fat content, with the 1.5% ES100 group having the highest moisture content (61.30 g/100 g). This is mainly due to the good water retention of the emulsion gel formed by thermal-denatured whey proteins, which prevents water loss during cheese processing [21]. The pre-emulsified fat could be evenly distributed in the casein network structure, which enhanced the connectivity of the three-dimensional network of proteins and its spatial hierarchy, thus improving the stability of water molecules in the cheese system [22]. In addition, it has been found that thermal-denatured whey proteins have the ability to bind water, and the addition of WPI to the milk base enhances the water-holding capacity of the system [23].

### 3.2. Effects on Brightness and Color

The color analysis of processed cheese samples is shown in Table 3. The difference in the L* parameter between the samples was significant. The L* value of 1% ES100 was lower (*p* < 0.05) than that of 1.5% ES100 and 3% ES100, and there was no significant difference (*p* > 0.05) between 1.5% ES100 and 3% ES100. Mozuraityte, Berget, Mahdalova, Grønsberg, Øye and Greiff [3] found that increasing emulsifying salt amount led to a slight increase in brightness. L* tended to increase with increasing degree of emulsification (1.5% ES control, 1.5% ES25, 1.5% ES50 and 1.5% ES100). The results showed that adding emulsion gel with a high degree of fat emulsification to processed cheese could increase the lightness of the cheese. Parameter a* gave positive values, indicating a tendency towards a red color. There was no significant difference (*p* > 0.05) in a* values between 1% ES100 and 1.5% ES100, and both were lower (*p* < 0.05) than 3% ES100. In the 1.5% ES group, the 1.5% ES control sample had the lowest (*p* < 0.05) a* values, and there was no significant difference (*p* > 0.05) in a* between 1.5% ES25, 1.5% ES50 and 1.5% ES100. The results showed that the content of pre-emulsified fat had little effect on a* values. Parameter b* gave positive values, indicating a tendency towards a yellow color [24]. The 3% ES100 group had the highest (*p* < 0.05) b* values. b* increased with the increased addition of emulsifying salt, while there was no significant difference (*p* > 0.05) in b* values between 1.5% ES25, 1.5% ES50 and 1.5% ES100.

### 3.3. Textural Properties

The TPA parameters of different cheese samples are shown in Table 4. The hardness values of 1% ES100, 1.5% ES100 and 3% ES100 were 145.86 g, 173.69 g and 318.41 g, respectively, which meant that the hardness values of the samples increased significantly with the increased addition of emulsifying salt. Many studies have confirmed that the addition of emulsifying salt changed the structure of casein, the structure of protein network became uniform and the intermolecular force of casein increased [12,25], which enhanced the strength of the network structure. In the 1.5% ES group (1.5% ES control, 1.5% ES25, 1.5% ES50 and 1.5% ES100), the hardness and adhesiveness increased significantly (*p* < 0.05) with the increase in emulsification degree of milk fat, but there was no significant difference (*p* > 0.05) in elasticity and chewiness between these samples. Among them, 1.5% ES100 showed the highest hardness and gumminess values with 173.69 g and 108.16 g, respectively. It indicated that the fat form in cheese played an important role in the texture of processed cheese. Fat can be used as a plasticizer in cheese and give it a softer texture [26], whereas pre-emulsified fat can give a firmer texture to the processed cheese. This was mainly due to the fact that as the content of pre-emulsified fat increased, there were more sites of interaction between thermal-denatured whey proteins and fats in the emulsified gel, and the gel structure was more compact, which enhanced the textural properties of the processed cheese product, and the hardness of the cheese was also higher. This result is consistent with previous studies [16,27]. There were no significant differences (*p* > 0.05) in textural properties between the 3% ES control group and the 1.5% ES100 group. Thermal-denatured whey proteins exposed more sulfhydryl and hydrophobic groups [28,29], and during heating, the generation of disulfide bonds and the enhancement of hydrophobicity promoted protein aggregation; meanwhile, casein and thermal-denatured whey proteins were crosslinked together [30]. As a result, it was easier to form systems with dense network structures. This suggests that pre-emulsification of fat with thermal-denatured whey proteins improves the emulsifying capacity of the samples, thus altering the texture of the samples and compensating for the deficiencies due to the reduction of emulsifying salts. This result was consistent with our previous study [16]. Moreover, thermal-denatured whey protein pre-emulsified 100% fat can reduce the amount of emulsifying salt added by 50%, reducing the sodium content of the cheese.

### 3.4. Melt and Free Oil Release Test

The results of meltability and free oil release of cheese samples are shown in Figure 1.

In the 1.5% ES cheese group, the extent of oil released significantly decreased with the increase in pre-emulsification degree. The oil released extent of 1.5% ES100 samples (2.1 cm) was reduced by 38% compared with the 1.5% ES control sample (3.4 cm). This result suggests that the formation of thermal-denatured whey protein emulsion gels improved the fat distribution of the samples and reduced the oil release. On the one hand, this may be due to the fact that the whey proteins encapsulated the fat to form an oil-in-water emulsion gel during the pre-emulsification process, which restricted the fat flow. On the other hand, the thermal-denatured whey proteins pre-emulsified the fat so that the fat was uniformly distributed in the cheese system in the form of small oil droplets, which enhanced the stability of the network structure [31]. Thus, emulsion gels made using the thermal-denatured whey protein pre-emulsification process can resist the problem of reduced emulsification due to the reduction of emulsifying salts. As is known, the processed cheese structure essentially consists of casein gel network and fat phase dispersed evenly in it [2]. This further suggests that the form of the fat is a key factor influencing the release of oil from the cheese. Meltability reflects the flow and spreading of cheese after heating. The 1.5% ES100 group had lower meltability compared to the 1.5% ES control (*p* < 0.05). During cheese preparation, thermal-denatured whey protein and κ-casein formed polymers through exchange interactions between sulfhydryl-disulfide bonds, leading to cross-linking of WPI with casein and an increase in internal forces, forming a dense network structure that limited intermolecular mobility of fats and proteins, thus decreasing the meltability of cheese samples [12,16,32]. In addition, there was a decreasing trend in the meltability and free oil release of the samples as the emulsifying salt content in the cheese increased (1% ES100, 1.5% ES100 and 3% ES100). Usually, the sodium ions in the emulsifying salt are exchanged with the calcium ions in the calcium casein phosphate, and the resulting sodium casein phosphate is attached to the surface of the fat globules, which serves to emulsify the fat in the cheese [33]. Therefore, as the content of emulsifying salts added increased, the emulsifying capacity of the cheese system was enhanced, and at the same time, it contributed to the expulsion of air from the cheese system. Then the macropores gradually disappeared or smaller pores were formed, and the structure of the cheese became continuous and dense, which resulted in a decrease in the degree of free oil release and meltability.

### 3.5. Confocal Laser Scanning Microscopy (CLAM)

Figure 2 shows the CLAM scans of different samples. In the figure, the fat phase is labeled in green and the protein phase in red. The 1% ES100 group and the 1.5% ES control group had obvious free fat aggregation and uneven fat distribution. The micrograph from CLSM could give a more intuitive indication of cheese meltability and free oil release conditions. The fat distribution of 1% ES100 and 1.5% ES control cheese groups was uneven, and there was obvious fat accumulation. With the increased extent of pre-emulsified process, the fat particle size of 1.5% ES25 and 1.5% ES50 cheese groups decreased significantly compared with the 1.5% ES control group, but there was still a certain degree of fat aggregation. In the 1.5% ES100, 3% ES control and 3% ES100 cheese groups, milk fat was evenly dispersed in the continuous protein matrix. Compared with the 3% ES control and 3% ES100 cheese groups, the amount of emulsifying salt in the 1.5% ES100 cheese group decreased by 50%. It can be concluded that pre-emulsification of milk fat could improve the functional properties (meltability and free oil release conditions) (Figure 1) and microstructure of reduced-sodium processed cheese. The 1.5% ES100 group with pre-emulsified 100% fat had a more continuous protein network compared to the 1.5% ES control cheese group.

In addition, with the increased extent of pre-emulsification (1.5% ES control, 1.5% ES25, 1.5% ES50 and 1.5% ES100) and addition of emulsifying salt (1% ES100, 1.5% ES100 and 3% ES100), the amounts of big free fat drops gradually decreased, and the fine spherical emulsified globular fat drops increased. This result was consistent with our previous study [16]. The reduction of emulsifying salt in processed cheese will lead to insufficient emulsification of fat and discontinuous distribution of protein networks. Therefore, how to form a uniform emulsion gel system in reduced-sodium cheese is a key problem to be solved [5,34,35]. According to our studies, the pre-emulsified process with thermal-denatured whey protein in reduced-sodium cheese resulted in a more uniform fat distribution and less fat release.

### 3.6. Sensory Analysis

Table 5 shows the average value of each sensory evaluation index. The average value of all products ranged from 5.67 to 7.76 (from I like it a little to I like it very much), indicating that the products were acceptable. The overall score for 3% ES100 was the highest, and 1.5% ES100 was close to it, indicating that the acceptability of the 1.5% ES100 sample was high.

Appearance scores increased with increasing emulsifying salt content (1% ES100, 1.5% ES100 and 3% ES100). Furthermore, in the 1.5% ES group (1.5% ES control, 1.5% ES25, 1.5% ES50 and 1.5% ES100), the higher the degree of pre-emulsification, the higher the score. This result was consistent with expectations.

With regard to color, the 3% ES100 group scored the highest (*p* < 0.05), showing the most natural orange color and shine. The 1.5% ES100 group scored slightly lower than the 3% ES100 group, but its acceptance was still good, whereas the 1% ES100 group scored noticeably lower, with a darker color. The results suggest that the emulsifying salts contributed to improving the color of the processed cheese [3].

The texture of 1% ES100 was uneven and slightly rough, while 1.5% ES100 and 3% ES100 had significantly better texture. Furthermore, the higher the degree of fat emulsification, the better the texture of the samples, indicating that the pre-emulsified fat is better encapsulated in the protein matrix to form a homogeneous state. The aroma showed the same trend. Milk fat decomposition is the main reason for the formation of the unique flavor of cheese [36]; in addition, thermal-denatured whey proteins can promote the binding and release of flavor substances [37]. Therefore, the thermal-denatured whey protein pre-emulsification process was effective in improving the organoleptic properties of reduced-sodium cheeses and enhancing their acceptability to consumers.

The hardness values of the 1% ES group were higher than those of the 1.5% ES group but were not significantly different from those of the 3% ES group. The hardness values of the 1.5% ES group increased with the increase in pre-emulsification degree of fat, which was in agreement with the results of the textural measurements (Table 4).

## 4. Conclusions

This study showed that the thermal-denatured whey protein pre-emulsification process resulted in a 50% reduction in the addition of emulsifying salts to the processed cheese while resolving the problem of lower quality of reduced-sodium cheeses due to the reduction of emulsifying salts. Thermal-denatured whey proteins encapsulated the free fat and enhanced the emulsification capacity of the cheese system, improving the fat distribution and fat globule size, which were evenly distributed in the casein network structure. In addition, as the pre-emulsified fat content increased, the protein cross-linking network became denser, enhancing the texture and reducing the meltability of the processed cheese. The results show that the thermal-denatured whey protein pre-emulsification process can improve emulsification during the processing of reduced-sodium processed cheeses and effectively contribute to the development of the cheese industry, especially in the children’s processed cheese market.

## Figures and Tables

**Figure 1 foods-12-02884-f001:**
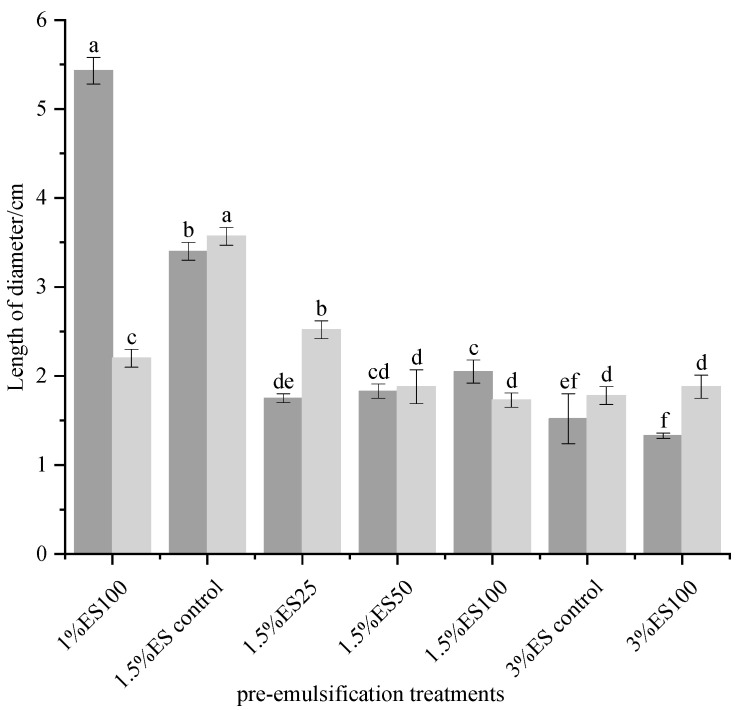
Free oil release and meltability of processed cheeses made from emulsion gels with different contents of emulsifying salts and different degrees of pre-emulsification. Free oil released (
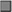
) and meltability (
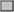
) of the different cheese samples. (1% ES100) = experimental group added with 1% emulsifying salt and pre-emulsified 100% fat with 5% WPI dispersion, (1.5% ES control) = control group added with 1.5% emulsifying salt, 5% WPI dispersion but not pre-emulsified fat, (1.5% ES25) = experimental group added with 1.5% emulsifying salt and pre-emulsified 25% fat with 5% WPI dispersion, (1.5% ES50) = experimental group added with 1.5% emulsifying salt and pre-emulsified 50% fat with 5% WPI dispersion, (1.5% ES100) = experimental group added with 1.5% emulsifying salt and pre-emulsified 100% fat with 5% WPI dispersion, (3% ES control) = control group added with 3% emulsifying salt, 5% WPI dispersion but not pre-emulsified fat, (3% ES100) = experimental group added with 3% emulsifying salt and pre-emulsified 100% fat with 5% WPI dispersion. ES: emulsifying salt, WPI: whey protein isolate. Means with different letters in the same column are significantly different by the Tukey test (*p* < 0.05).

**Figure 2 foods-12-02884-f002:**
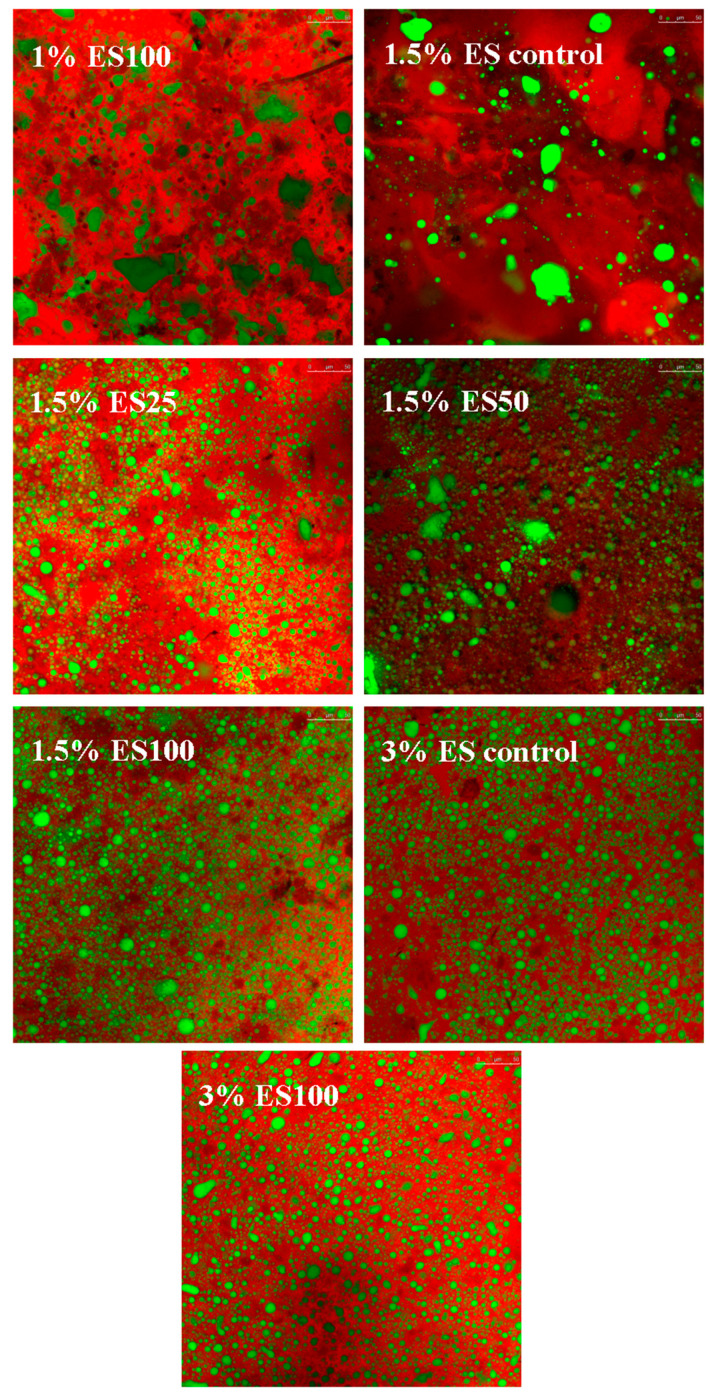
Confocal laser scanning microscopy (CLSM) of processed cheeses with different degrees of pre-emulsification and different contents of emulsifying salts. (1% ES100) = experimental group added with 1% emulsifying salt and pre-emulsified 100% fat with 5% WPI dispersion, (1.5% ES control) = control group added with 1.5% emulsifying salt, 5% WPI dispersion but not pre-emulsified fat, (1.5% ES25) = experimental group added with 1.5% emulsifying salt and pre-emulsified 25% fat with 5% WPI dispersion, (1.5% ES50) = experimental group added with 1.5% emulsifying salt and pre-emulsified 50% fat with 5% WPI dispersion, (1.5% ES100) = experimental group added with 1.5% emulsifying salt and pre-emulsified 100% fat with 5% WPI dispersion, (3% ES control) = control group added with 3% emulsifying salt, 5% WPI dispersion but not pre-emulsified fat, (3% ES100) = experimental group added with 3% emulsifying salt and pre-emulsified 100% fat with 5% WPI dispersion. ES: emulsifying salt, WPI: whey protein isolate.

**Table 1 foods-12-02884-t001:** Formulation of processed cheeses with different degrees of pre-emulsification and different contents of emulsifying salts.

Sample ^1^	Cheese (g)	WPI-EGs(WPI/g Water/g AMF/g)	AMF (g)	Casein (g)	Emulsifying Salts (g)	Water (g)
1% ES100	600	840 (29, 551, 260)	0	200	20	340
1.5% ES control	600	0 (29, 551, 260)	0	200	30	330
1.5% ES25	600	645 (29, 551, 65)	195	200	30	330
1.5% ES50	600	710 (29, 551, 130)	130	200	30	330
1.5% ES100	600	840 (29, 551, 260)	0	200	30	330
3% ES control	600	0 (29, 551, 260)	0	200	60	300
3% ES100	600	840 (29, 551, 260)	0	200	60	300

Emulsifying salts were composed of sodium citrate and sodium pyrophosphate in the ratio of 6.5:1. ^1^ (1% ES100) = experimental group added with 1% emulsifying salt and pre-emulsified 100% fat with 5% WPI dispersion, (1.5% ES control) = control group added with 1.5% emulsifying salt, 5% WPI dispersion but not pre-emulsified fat, (1.5% ES25) = experimental group added with 1.5% emulsifying salt and pre-emulsified 25% fat with 5% WPI dispersion, (1.5% ES50) = experimental group added with 1.5% emulsifying salt and pre-emulsified 50% fat with 5% WPI dispersion, (1.5% ES100) = experimental group added with 1.5% emulsifying salt and pre-emulsified 100% fat with 5% WPI dispersion, (3% ES control) = control group added with 3% emulsifying salt, 5% WPI dispersion but not pre-emulsified fat, (3% ES100) = experimental group added with 3% emulsifying salt and pre-emulsified 100% fat with 5% WPI dispersion. ES: emulsifying salt, WPI: whey protein isolate, AMF: anhydrous milk fat, EGs: emulsion gels.

**Table 2 foods-12-02884-t002:** Physicochemical composition of processed cheeses with different degrees of pre-emulsification and different contents of emulsifying salts.

Sample ^1^	Protein (g/100 g)	Fat (g/100 g)	Moisture (g/100 g)	Ash (g/100 g)	pH	Na (mg/100 g)
1% ES100	16.16 ± 0.58 ^a^	20.48 ± 0.52 ^a^	59.76 ± 0.31 ^d^	3.56 ± 0.05 ^a^	6.07 ± 0.02 ^a^	271.06 ± 15.57 ^a^
1.5% ES control	16.47 ± 0.58 ^a^	19.91 ± 0.48 ^a^	57.74 ± 0.36 ^c^	4.34 ± 0.03 ^b^	6.08 ± 0.04 ^a^	517.06 ± 4.28 ^b^
1.5% ES25	16.29 ± 0.69 ^a^	20.33 ± 0.57 ^a^	57.80 ± 0.17 ^c^	4.40 ± 0.03 ^b^	6.14 ± 0.01 ^b^	522.40 ± 7.83 ^b^
1.5% ES50	16.00 ± 0.55 ^a^	19.97 ± 0.47 ^a^	59.59 ± 0.69 ^d^	4.33 ± 0.03 ^b^	6.10 ± 0.04 ^ab^	531.69 ± 7.23 ^b^
1.5% ES100	16.21 ± 0.24 ^a^	20.92 ± 0.38 ^a^	61.30 ± 0.22 ^e^	4.41 ± 0.03 ^b^	6.12 ± 0.05 ^ab^	535.97 ± 15.34 ^b^
3% ES control	16.39 ± 0.01 ^a^	20.01 ± 0.56 ^a^	53.30 ± 0.19 ^a^	5.28 ± 0.03 ^c^	6.37 ± 0.04 ^c^	936.29 ± 11.82 ^c^
3% ES100	16.38 ± 0.51^a^	20.69 ± 0.62 ^a^	55.31 ± 0.23 ^b^	5.21 ± 0.07 ^c^	6.42 ± 0.06 ^c^	947.08 ± 13.03 ^c^

All values are mean ± s.d. Values in the same column with different superscript letters are significantly different (*p* < 0.05). ^1^ (1% ES100) = experimental group added with 1% emulsifying salt and pre-emulsified 100% fat with 5% WPI dispersion, (1.5% ES control) = control group added with 1.5% emulsifying salt, 5% WPI dispersion but not pre-emulsified fat, (1.5% ES25) = experimental group added with 1.5% emulsifying salt and pre-emulsified 25% fat with 5% WPI dispersion, (1.5% ES50) = experimental group added with 1.5% emulsifying salt and pre-emulsified 50% fat with 5% WPI dispersion, (1.5% ES100) = experimental group added with 1.5% emulsifying salt and pre-emulsified 100% fat with 5% WPI dispersion, (3% ES control) = control group added with 3% emulsifying salt, 5% WPI dispersion but not pre-emulsified fat, (3% ES100) = experimental group added with 3% emulsifying salt and pre-emulsified 100% fat with 5% WPI dispersion. ES: emulsifying salt, WPI: whey protein isolate.

**Table 3 foods-12-02884-t003:** Color analysis of processed cheeses with different degrees of pre-emulsification and different contents of emulsifying salts.

Samples ^1^	1% ES100	1.5% ES Control	1.5% ES25	1.5% ES50	1.5% ES100	3% ES control	3% ES100
L*	75.54 ± 0.66 ^b^	73.33 ± 0.87 ^a^	77.81 ± 0.57 ^c^	79.95 ± 0.83 ^d^	79.37 ± 0.76 ^d^	79.96 ± 1.02 ^d^	79.88 ± 0.76 ^d^
a*	9.94 ± 0.35b ^b^	8.96 ± 0.35 ^a^	9.71 ± 0.08 ^b^	9.98 ± 0.02 ^b^	10.00 ± 0.29 ^b^	10.14 ± 0.06 ^b^	11.13 ± 0.16 ^c^
b*	30.80 ± 0.49 ^b^	33.42 ± 0.42 ^c^	28.48 ± 0.046 ^a^	28.99 ± 0.44 ^a^	28.62 ± 0.48 ^a^	31.69 ± 0.14 ^c^	34.88 ± 0.40 ^d^

All values are mean ± s.d. Values in the same column with different superscript letters are significantly different (*p* < 0.05). ^1^ (1% ES100) = experimental group added with 1% emulsifying salt and pre-emulsified 100% fat with 5% WPI dispersion, (1.5% ES control) = control group added with 1.5% emulsifying salt, 5% WPI dispersion but not pre-emulsified fat, (1.5% ES25) = experimental group added with 1.5% emulsifying salt and pre-emulsified 25% fat with 5% WPI dispersion, (1.5% ES50) = experimental group added with 1.5% emulsifying salt and pre-emulsified 50% fat with 5% WPI dispersion, (1.5% ES100) = experimental group added with 1.5% emulsifying salt and pre-emulsified 100% fat with 5% WPI dispersion, (3% ES control) = control group added with 3% emulsifying salt, 5% WPI dispersion but not pre-emulsified fat, (3% ES100) = experimental group added with 3% emulsifying salt and pre-emulsified 100% fat with 5% WPI dispersion. ES: emulsifying salt, WPI: whey protein isolate.

**Table 4 foods-12-02884-t004:** TPA parameters of processed cheeses with different degrees of pre-emulsification and different contents of emulsifying salts.

Sample ^1^	Hardness (g)	Springiness	Cohesiveness	Gumminess(N)	Chewiness (g)
1% ES100	145.86 ± 26.49 ^cd^	0.67 ± 0.02 ^b^	0.49 ± 0.02 ^b^	70.78 ± 10.58 ^d^	47.25 ± 5.63 ^d^
1.5% ES control	98.58 ± 19.98 ^e^	0.71 ± 0.01 ^ab^	0.50 ± 0.04 ^b^	49.75 ± 14.19 ^e^	35.23 ± 9.63 ^d^
1.5% ES25	128.48 ± 1.91 ^d^	0.71 ± 0.04 ^ab^	0.58 ± 0.11 ^ab^	73.95 ± 15.06 ^d^	52.77 ± 13.46 ^d^
1.5% ES50	149.38 ± 4.36 ^cd^	0.75 ± 0.02 ^a^	0.64 ± 0.02 ^a^	94.85 ± 5.37 ^c^	70.95 ± 5.78 ^c^
1.5% ES100	173.69 ± 4.76 ^bc^	0.76 ± 0.03 ^a^	0.62 ± 0.02 ^a^	108.16 ± 6.10 ^bc^	82.55 ± 7.96 ^bc^
3% ES control	183.43 ± 6.58 ^b^	0.77 ± 0.03 ^a^	0.65 ± 0.01 ^a^	118.69 ± 4.93 ^b^	91.17 ± 0.48 ^b^
3% ES100	300.98 ± 7.28 ^a^	0.74 ± 0.02 ^a^	0.59 ± 0.02 ^ab^	178.76 ± 0.17 ^a^	132.61 ± 3.89 ^a^

All values are mean ± s.d. Values in the same column with different superscript letters are significantly different (*p* < 0.05). ^1^ (1% ES100)= experimental group added with 1% emulsifying salt and pre-emulsified 100% fat with 5% WPI dispersion, (1.5% ES control) = control group added with 1.5% emulsifying salt, 5% WPI dispersion but not pre-emulsified fat, (1.5% ES25) = experimental group added with 1.5% emulsifying salt and pre-emulsified 25% fat with 5% WPI dispersion, (1.5% ES50) = experimental group added with 1.5% emulsifying salt and pre-emulsified 50% fat with 5% WPI dispersion, (1.5% ES100) = experimental group added with 1.5% emulsifying salt and pre-emulsified 100% fat with 5% WPI dispersion, (3% ES control) = control group added with 3% emulsifying salt, 5% WPI dispersion but not pre-emulsified fat, (3% ES100) = experimental group added with 3% emulsifying salt and pre-emulsified 100% fat with 5% WPI dispersion. ES: emulsifying salt, WPI: whey protein isolate.

**Table 5 foods-12-02884-t005:** Mean scores for the attributes of appearance, color, texture, aroma and firmness for the sample of 1%ES100, 1.5%ES group and 3%ES group.

Sample ^1^	Appearance	Color	Texture	Aroma	Firmness
1% ES100	6.00 ^a^	6.13 ^a^	5.67 ^a^	6.57 ^a^	7.10 ^cd^
1.5% ES control	6.00 ^a^	6.57 ^b^	5.73 ^a^	6.70 ^ab^	6.37 ^a^
1.5% ES25	6.33 ^ab^	6.60 ^b^	5.80 ^a^	6.87 ^abc^	6.43 ^a^
1.5% ES50	6.80 ^bc^	6.83 ^bc^	6.27 ^b^	6.93 ^abc^	6.50 ^ab^
1.5% ES100	7.23 ^cd^	6.87 ^bc^	6.30 ^b^	7.00 ^bc^	6.87 ^bc^
3% ES control	7.40 ^cd^	6.97 ^c^	6.50 ^c^	7.13 ^c^	7.37 ^d^
3% ES100	7.76 ^d^	7.07 ^c^	6.63 ^c^	7.23 ^c^	7.37 ^d^

Means with different letters in the same column are significantly different by the Tukey test (*p* < 0.05). ^1^ (1% ES100) = experimental group added with 1% emulsifying salt and pre-emulsified 100% fat with 5% WPI dispersion, (1.5% ES control) = control group added with 1.5% emulsifying salt, 5% WPI dispersion but not pre-emulsified fat, (1.5% ES25) = experimental group added with 1.5% emulsifying salt and pre-emulsified 25% fat with 5% WPI dispersion, (1.5% ES50) = experimental group added with 1.5% emulsifying salt and pre-emulsified 50% fat with 5% WPI dispersion, (1.5% ES100) = experimental group added with 1.5% emulsifying salt and pre-emulsified 100% fat with 5% WPI dispersion, (3% ES control) = control group added with 3% emulsifying salt, 5% WPI dispersion but not pre-emulsified fat, (3% ES100) = experimental group added with 3% emulsifying salt and pre-emulsified 100% fat with 5% WPI dispersion. ES: emulsifying salt, WPI: whey protein isolate.

## Data Availability

The data presented in this study are available on request from the corresponding author. The data are not publicly available due to [privacy].

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
