# Peer review of "Effects of Pre-Emulsification with Thermal-Denatured Whey Protein on Texture and Microstructure of Reduced-Sodium Processed Cheese"

_foods, 2023, doi:10.3390/foods12152884_

Round 1

Reviewer 1 Report

This manuscript is about effects of pre-emulsification with thermal-denatured whey protein on texture and microstructure of reduced-sodium processed cheese. It is interesting and I think. You can find my comments in below:

1. The manuscript must be revised grammatically and the English level of it must be improved by a native editor.

2. The authors must re-write the abstract and conclusion sections. I think some sentences are not needed to be in these sections.

3. In introduction section, at first please add some information about processed cheese and its production.

4. In introduction section, please add more references about the same researches which worked on current research title.

5. Please re-write section 2.5. It is not that much clear.

6. In section results and discussion, please give more references (like https://doi.org/10.1016/j.foodhyd.2022.107758) and details. Also, compare the results with the results obtained by previous researches.  

7. Please increase the DPI values of figures. The quality of them is poor. Also, please make the tables more desirable. They are not looking that much good.   

The manuscript must be revised grammatically and the English level of it must be improved by a native editor

Author Response

Response to Reviewer 1 Comments

Dear Reviewer,

Thank you very much for the critical comments and valuable suggestions on our manuscript (foods-2504855). We have revised the paper and would like to re-submit it for your consideration. The amendments are highlighted in yellow in the revised manuscript. We hope that the revision is acceptable, and I look forward to hearing from you. Below you will find our point-by-point responses to the reviewer's comments/questions.

1. The manuscript must be revised grammatically and the English level of it must be improved by a native editor.

Answer: We sincerely thank the editor and all reviewers for their valuable feedback, which we have used to improve the quality of our manuscript.

We have performed English proofreading.

2. The authors must re-write the abstract and conclusion sections. I think some sentences are not needed to be in these sections.

Answer: Thanks to the reviewer's valuable suggestions. We have rewritten the sections “Abstract” and “Conclusion”.

3. In introduction section, at first please add some information about processed cheese and its production.

Answer: Thanks to the reviewer's valuable suggestions. We have added some information about processed cheese and its production.

Processed cheese, which is rich in nutritional value, is a homogeneous product formed by heating and continuous mixing of one or two natural cheeses, water, emulsifying salts and other dairy or non-dairy products. It has a high level of acceptance due to its wide range of sources, low fermentation flavor and long shelf life.

4. In introduction section, please add more references about the same researches which worked on current research title.

Answer: Thanks to the reviewer's valuable suggestions. We have added more references about the same researches which worked on current research title.

5. Please re-write section 2.5. It is not that much clear.

Answer: Thanks to the reviewer's valuable suggestions. We have rewritten the section 2.5.

“The method of Li, Qin, Yu, Han, Zheng, Li and Yu [16] was referred to with some modifications. A 15 x 15 x 15 mm cube sample was cut from the center of the processed cheese. TPA analyses were performed on cheeses after 2 weeks of storage at 4°C using a TA-XT2i (Stable Micro Systems, Surrey, UK). The parameters of texture analyzer were set as follows: before the test, the probe descending speed was 4.0 mm/s; the test speed was 1.0 mm/s; after the test, the probe return speed was 1.0 mm/s; the compression deformation was 30%; the trigger force was 10 g; and the test was performed using a cylindrical metal compression plate of 36 mm diameter. All tests were performed at room temperature, and all analyses were performed in triplicate.”

6. In section results and discussion, please give more references (like https://doi.org/10.1016/j.foodhyd.2022.107758) and details. Also, compare the results with the results obtained by previous researches.  

Answer: Thanks to the reviewers for their valuable suggestions. We have added more relevant references to support my conclusions in the discussion section of the manuscript.

7. Please increase the DPI values of figures. The quality of them is poor. Also, please make the tables more desirable. They are not looking that much good.   

Answer: Thanks to the reviewers for their valuable suggestions. Because the font in the chart is size 10, it presents such clarity. If you enlarge the entire Word page, the clarity improves.

Comments on the Quality of English Language

The manuscript must be revised grammatically and the English level of it must be improved by a native editor

Answer: We sincerely thank the editor and all reviewers for their valuable feedback, which we have used to improve the quality of our manuscript.

We have performed English proofreading.

Reviewer 2 Report

This study aimed to develop a processed cheese with a reduced sodium content. The effect of pre-emulsification with thermal-denatured whey protein on texture and microstructure of reduced-sodium processed cheese was studied.  The study needs more justifications and revision in order to improve the current quality, my recommendations are listed below;

-Abstract should be rewritten and more quantitative (numerical) findings should be added.

-Write briefly and clearly in abstract the pre-emulsification with thermal-denatured whey protein, how you condcuted that, as this apporach is the main way to achieve the aims. For example mention the several treatments of emulsifying salts added... 

-L 59: convert The to the

-L 85: Check this sentence, confusing? used polymerised whey protein isolate... Who is used?

-L 103: write in more details this approach (Pre-emulsified processing was prepared as previously reported methods...).

-L 104: write the model of the oven used as well its country etc. Apply this issue for all instruments used in the study.

-L 104; add the stirring time.

- L 122, convert rpm to xg, add the centrifugation time.

-L 166; add the manufacturer and country of SPSS software.

-Support your discussion with more relevant reports, in particularly sensory evaluation.

-Mention in the sensory method the sensorial attributes that you carried out (Appearance, Color, Texture, Aroma, Firmness), my comment here is Why you didnt conduct the taste attribute of the treatments??

- Double check the references as the journal guide.

-Reference 27: remove 1 after cream.

- Edit the title of Table 1.

-Specify the complete meaning of each abbreviated word in the tables and figures like AMF, in the table footnote/ or figure caption....

-Extend the title of all tables to express the scientific content of the table, like you could write the treatments condcuted etc.

-The title of Fig. 1 is non clear?¿

- Fig. 1: write under X axis (pre-emulsification treatments).

Must be improved.

Author Response

Response to Reviewer 2 Comments

Dear Reviewer,

Thank you very much for the critical comments and valuable suggestions on our manuscript (foods-2504855). We have revised the paper and would like to re-submit it for your consideration. The amendments are highlighted in yellow in the revised manuscript. We hope that the revision is acceptable, and I look forward to hearing from you. Below you will find our point-by-point responses to the reviewer's comments/questions.

-Abstract should be rewritten and more quantitative (numerical) findings should be added.

-Write briefly and clearly in abstract the pre-emulsification with thermal-denatured whey protein, how you condcuted that, as this apporach is the main way to achieve the aims. For example mention the several treatments of emulsifying salts added...

Answer: Thanks to the reviewer's valuable suggestions. We have modified the section “Abstract”.

We have added the following to the section “abstract”: Thermal-denatured whey protein-milk fat emulsion gels with different degrees of pre-emulsification were prepared by pre-emulsifying milk fat with thermal-denatured whey protein and used in the preparation of reduced-sodium processed cheeses.

-L 59: convert The to the

Answer: Thanks to the reviewer's valuable suggestions.

However, as this sentence is not quite appropriate here, we have removed it.

-L 85: Check this sentence, confusing? used polymerised whey protein isolate... Who is used?

Answer: Thanks to the reviewer's valuable suggestions. We have rewritten the sentence.

Sołowiej, et al. [12] used polymerized whey protein isolate (0.2% to 0.4% WPI) as a partial replacement for emulsifying salt in the production of processed cheese. It was found that the tested samples all had good meltability.

-L 103: write in more details this approach (Pre-emulsified processing was prepared as previously reported methods...).

Answer: Thanks to the reviewer's valuable suggestions. We have rewritten the section “2.2.1 Pre-emulsification of milk fat with thermal-denatured whey protein”.

“It was prepared according to the previously reported method [17]. WPI was dissolved in distilled water and rapidly stirred with a high-speed disperser (FJ200-SH, Shanghai Standard Model Co., Ltd., Shanghai, China) at 9000 rpm for 3 min at room temperature, followed by hydration in a water bath at 55 °C for 1 h, and finally denaturation at 90 °C for 20 min to obtain WPI dispersions with a mass fraction of 5%. Then, milk fat (60 °C) was added to the WPI solution and stirred rapidly at 10,000 rpm for 3 min with a high-speed disperser. After cooling to room temperature, the pH of the emulsion was adjusted with citric acid to 6.0. After storage at 4 °C for 12 h, thermal-denatured whey protein-milk fat emulsion gels (WPI-EGs) were formed.”

-L 104: write the model of the oven used as well its country etc. Apply this issue for all instruments used in the study.

Answer: Thanks to the reviewer's valuable suggestions. However, we did not use an oven in the experiment. We have checked and corrected all the information on the country and model of the equipment in the manuscript.

-L 104; add the stirring time.

Answer: Thanks to the reviewer's valuable suggestions. We have added the stirring time: 3min.

- L 122, convert rpm to xg, add the centrifugation time.

Answer: Thanks to the reviewer's valuable suggestions. We have changed 250 rpm to 300 r/min and the stirring time to 40min. Since the equipment used is a vacuum homogenizing emulsifying machine, the unit is: r/min.

-L 166; add the manufacturer and country of SPSS software.

Answer: Thanks to the reviewer's valuable suggestions. We have added the manufacturer and country of SPSS software.

SPSS 13.0 (SPSS Inc., Chicago, IL, USA).

-Support your discussion with more relevant reports, in particularly sensory evaluation.

Answer: Thanks to the reviewers for their valuable suggestions. We have added more relevant reports to support my conclusions in the discussion section of the manuscript.

-Mention in the sensory method the sensorial attributes that you carried out (Appearance, Color, Texture, Aroma, Firmness), my comment here is Why you didnt conduct the taste attribute of the treatments??

Answer: Thanks to the reviewers for their valuable suggestions. We tested the taste attributes in the pre-experiment and found that there was a slight difference in savory flavor between the groups of samples, but the difference was not significant. In contrast, there were significant differences in appearance, color, texture, aroma and firmness, so ultimately taste attributes were not included in the final sensory methods.

- Double check the references as the journal guide.

Answer: Thanks to the reviewer's valuable suggestions. We have carefully checked the references.

-Reference 27: remove 1 after cream.

Answer: Thanks to the reviewer's valuable suggestions. We have removed 1 after cream.

- Edit the title of Table 1.

Answer: Thanks to the reviewer's valuable suggestions. We have rewritten the title of Table 1: “Formulation of processed cheeses with different degrees of pre-emulsification and different contents of emulsifying salts".

-Specify the complete meaning of each abbreviated word in the tables and figures like AMF, in the table footnote/ or figure caption....

Answer: Thanks to the reviewer's valuable suggestions. We have checked and supplemented the complete meanings of each abbreviated word in tables and figures.

ES: emulsifying salt, WPI: Whey protein isolate, AMF: Anhydrous milk fat, EGs: Emulsion gels.

-Extend the title of all tables to express the scientific content of the table, like you could write the treatments condcuted etc.

Answer: Thanks to the reviewer's valuable suggestions. We have rewritten the title of tables1-4.

Table 1: Formulation of processed cheeses with different degrees of pre-emulsification and different contents of emulsifying salts.

Table 2: Physicochemical composition of processed cheeses with different degrees of pre-emulsification and different contents of emulsifying salts.

Table 3 Color analysis of processed cheeses with different degrees of pre-emulsification and different contents of emulsifying salts.

Table 4 TPA parameters of processed cheeses with different degrees of pre-emulsification and different contents of emulsifying salts.

-The title of Fig. 1 is non clear?¿

Answer: Thanks to the reviewer's valuable suggestions. We have rewritten the title of Fig. 1 “Free oil release and meltability of processed cheeses with different degrees of pre-emulsification and different contents of emulsifying salts.”.

- Fig. 1: write under X axis (pre-emulsification treatments).

Answer: Thanks to the reviewer's valuable suggestions. We have added “pre-emulsification treatments” under X axis in Fig 1.

Comments on the Quality of English Language

Must be improved.

Answer: We sincerely thank the editor and all reviewers for their valuable feedback, which we have used to improve the quality of our manuscript.

We have performed English proofreading.

Round 2

Reviewer 2 Report

-Define the abbreviation mean first time mentioned (ES in line 31).

-Line 134: convert (1.5%EScontrol) to (1.5% ES control) apply this issue for the many similar cases in the manuscript.

Fine, and can be improved